# Perception of Biocontrol Potential of *Bacillus*
*inaquosorum* KR2-7 against Tomato Fusarium Wilt through Merging Genome Mining with Chemical Analysis

**DOI:** 10.3390/biology11010137

**Published:** 2022-01-14

**Authors:** Maedeh Kamali, Dianjing Guo, Shahram Naeimi, Jafar Ahmadi

**Affiliations:** 1College of Veterinary Medicine and Life Sciences, City University of Hong Kong, Hong Kong, China; mkamali@cityu.edu.hk; 2State Key Laboratory of Agrobiotechnology and School of Life Sciences, Chinese University of Hong Kong, Hong Kong, China; 3Department of Biological Control Research, Iranian Research Institute of Plant Protection, Agricultural Research, Education and Extension Organization (AREEO), Tehran 19858-13111, Iran; sh.naeimi@areeo.ac.ir; 4Department of Genetics and Plant Breeding, Imam Khomeini International University, Qazvin 34149-16818, Iran; j.ahmadi@eng.ikiu.ac.ir

**Keywords:** Fusarium wilt, biocontrol, *B. inaquosorum* KR2-7, genome mining, gene clusters, MALDI-TOF-MS, secondary metabolites

## Abstract

**Simple Summary:**

*Bacillus* is a bacterial genus that is widely used as a promising alternative to chemical pesticides due to its protective activity toward economically important plant pathogens. Fusarium wilt of tomato is a serious fungal disease limiting tomato production worldwide. Recently, the newly isolated *B. inaquosorum* strain KR2-7 considerably suppressed Fusarium wilt of tomato plants. The present study was performed to perceive potential direct and indirect biocontrol mechanisms implemented by KR2-7 against this disease through genome and chemical analysis. The potential direct biocontrol mechanisms of KR2-7 were determined through the identification of genes involved in the synthesis of antibiotically active compounds suppressing tomato Fusarium wilt. Furthermore, the indirect mechanisms of this bacterium were perceived through recognizing genes that contributed to the resource acquisition or modulation of plant hormone levels. This is the first study that aimed at the modes of actions of *B. inaquosorum* against Fusarium wilt of tomatoes and the results strongly indicate that strain KR2-7 could be a good candidate for microbial biopesticide formulations to be used for biological control of plant diseases and plant growth promotion.

**Abstract:**

Tomato Fusarium wilt, caused by *Fusarium oxysporum* f. sp. *lycopersici* (*Fol*), is a destructive disease that threatens the agricultural production of tomatoes. In the present study, the biocontrol potential of strain KR2-7 against *Fol* was investigated through integrated genome mining and chemical analysis. Strain KR2-7 was identified as *B. inaquosorum* based on phylogenetic analysis. Through the genome mining of strain KR2-7, we identified nine antifungal and antibacterial compound biosynthetic gene clusters (BGCs) including fengycin, surfactin and Bacillomycin F, bacillaene, macrolactin, sporulation killing factor (skf), subtilosin A, bacilysin, and bacillibactin. The corresponding compounds were confirmed through MALDI-TOF-MS chemical analysis. The gene/gene clusters involved in plant colonization, plant growth promotion, and induced systemic resistance were also identified in the KR2-7 genome, and their related secondary metabolites were detected. In light of these results, the biocontrol potential of strain KR2-7 against tomato Fusarium wilt was identified. This study highlights the potential to use strain KR2-7 as a plant-growth promotion agent.

## 1. Introduction

Tomato Fusarium wilt, caused by *Fusarium oxysporum* f. sp. *lycopersici* (*Fol*), is one of the most destructive diseases, causing a considerable loss in the production of both field and greenhouse tomatoes worldwide [1]. Since Fusarium wilt is a difficult disease to control [2,3,4], control strategies including physical and cultural methods, chemical fungicides treatment, and the cultivation of resistance tomato cultivars [5] achieved limited efficacy [6]. In addition, excessive usage of agrochemicals imposed serious negative impacts on the environment, causing the pollution of soil and groundwater reservoirs, an accumulation of chemical residues in the food chain, an emergence of pesticide-resistance pathogens, and health hazards [7]. As a result, biocontrol microbes have been suggested as a promising alternative to agrochemicals in plant disease control. Numerous biocontrol microbes, especially *Bacillus* strains, have been commercially developed as biopesticides and biofertilizers worldwide [7].

Biocontrol microbes protect the crops from an invasion of phytopathogens via (1) direct modes of action, e.g., the antibiosis and production of antimicrobial secondary metabolites [8,9]; and (2), the indirect modes of action, including induced systemic resistance (ISR) and the competition for nutrients and space [10,11]. The investigation of biocontrol microbes through conventional genetic and biochemical approaches could not unveil the full potential of these microbes due to the absence of appropriate natural triggers or stress signals under laboratory conditions [12]. With the development of high-throughput DNA sequencing technologies and genome mining, along with MS-based analytical methods (e.g., GC/LC-MS, LC-ESI-MS, and MALDI-TOF-MS), more potential biocontrol microbes can be revealed. For instance, the *Bacillus amyloliquefaciens* FZB42 genome contains nine giant gene clusters synthesizing secondary metabolites which are involved in the suppression of soil-born plant pathogens. Several gene/gene clusters are implicated in swarming motility, plant colonization, biofilm formation, and the synthesis of plant growth-promoting volatile compounds and hormones [13]. A wide range of extracellular proteins and phytase in the FZB42 secretome were detected through two-dimensional electrophoresis, MALDI-TOF-MS, and the proteomics approach, indicating this strain can grow on the plant’s surface and supply phosphorus for the plant under phosphorus starvation. Additionally, four members of the macrolactin family were identified in an FZB42 culture filtrate by combining mass spectrometric and ultraviolet-visible data which perfectly agree with the overall structure of the macrolactin gene cluster found in the FZB42 genome [13]. Recently, the genome analysis of plant-protecting bacterium *B. velezensis* 9D-6 demonstrated that this strain can synthesize 13 secondary metabolites, of which surfacin B and surfactin C were detected as antimicrobial compounds against *Clavibacter michiganensis* through LC-MS/MS [14]. Furthermore, the genome mining of *B. inaquosorum* strain HU Biol-II revealed that this bacterial genome contains eight bioactive metabolite clusters and the production of seven metabolites was confirmed through HPLC MS/MS [15].

In our previous study, the *B. inaquosorum* strain KR2-7 was isolated from the rhizosphere soil of the tomato (*Solanum lycopersicum*) and was introduced as a highly effective biocontrol agent against *Fol* with a biocontrol efficiency of 80% under greenhouse conditions [16]. To better understand the biocontrol mechanisms of strain KR2-7 against *Fol*, whole-genome sequencing was conducted to identify putative gene clusters for secondary metabolites biosynthesis and to characterize gene/gene clusters involved in plant colonization, plant growth promotion, and induced systemic resistance (ISR). Moreover, secondary metabolites and other compounds related to identified BGCs and gene/gene clusters were detected using MALDI-TOF-MS analysis to confirm the results of genome mining.

## 2. Materials and Methods

### 2.1. Strains and Culture Conditions

The fungal pathogen *Fol* strain Fo-To-S-V-1 used in this study was obtained from the culture collection from the Iranian Research Institute of Plant Protection. The fungus was maintained on a potato dextrose agar (PDA, Merck, Germany) slant at 4 °C and was sub-cultured onto a fresh PDA plate at 27 °C for 7 days for further tests.

Strain KR2-7 was maintained on nutrient agar (NA, Merck, Germany; with a 0.3% beef extract, 0.5% peptone, and 1.5% agar) plate with a periodic transfer to a fresh medium. For long-term storage, it was kept at −80 °C in lysogeny broth (LB, Merck, Germany) with 20% glycerol (*v/v*).

### 2.2. Dual Culture Assay

In order to investigate the antagonism efficiency of strain KR2-7 against various tomato pathogens, five destructive fungal pathogens, including *Alternaria alternata* f. sp. *lycopersici*, *Athelia rolfsii*, *Botrytis cinerea*, *Rhizoctonia solani*, and *Verticillium albo-atrum*, were selected. The antifungal activity of strain KR2-7 against each pathogen was evaluated through a dual culture assay in three replications. In the dual culture assay, strain KR2-7 was simultaneously cultured 3cm apart from the 5-mm plug of a pathogen in a 9 cm PDA plate. The control plate was inoculated only with the pathogen. Plates were incubated at 27 °C. The fungal growth was checked daily by measuring the diameter of the colony for a period of three days. The percentage of fungal growth inhibition (PFGI) was calculated by the formula (1) developed by Skidmore and Dickinson [17], where R1 is the maximum radius of the growing fungal colony in the control plate, and R2 is the radius of the fungal colony that grew in the presence of strain KR2-7:PFGI = ((R_1_ − R_2_)/R_1_) × 100(1)

### 2.3. MALDI-TOF-MS Analysis of KR2-7 Secondary Metabolites

The secondary metabolite analysis was performed from the whole-cell surface extract of bacterium obtained during the dual culture of KR2-7 and *Fol*. The bacterial surface extract was prepared according to the methodology described by Vater et al. [18]. The Dual culture was done on potato dextrose agar (PDA) instead of Landy agar. Strain KR2-7 was streaked on one side of the plate and a 5 mm plug of *Fol* was placed on the opposite side simultaneously and incubated at 27 °C. After 24 h, two loops of bacterial cells from the interface of the bacterium-fungus in the inhibition zone were suspended in 500 µL of 70% acetonitrile with 0.1% trifluoroacetic acid for 2 min. The suspension was gently vortexed to produce a homogenized suspension. The bacterial cells were pelleted by centrifuging at 5000 rpm for 10 min. The cell-free supernatant was transferred to a new microcentrifuge tube and stored at 4 °C for further analysis. One microliter of supernatant liquid was spotted onto the target of the mass spectrometer with an equal volume of α-cyano-4-hydroxycinnamic acid (CHCA) matrix and was air-dried. The sample mass fingerprints were obtained using an ultrafleXtreme MALDI-TOF/TOF-MS (Bruker, Billerica, MA, USA) within a mass range of 100–3000 Da. The MALDI-TOF-MS analysis was performed at the school of life sciences, Chinese University of Hong Kong (CUHK), Hong Kong. The whole-cell surface extract of strain KR2-7 grown on a potato dextrose agar was used as a control. 

### 2.4. Genome Sequencing, Assembly, and Annotation

The genomic DNA of strain KR2-7 was extracted using a commercial DNA extraction kit (Thermo Fisher Scientific, Waltham, MA, USA). The whole-genome sequencing was performed using the Illumina HiSeq 4000 and PacBio RSII platforms (BGI, Shenzhen, China). The quality control of raw sequences was performed by FastQC v0.11.9 and the de novo assembly was done using SPAdes v3.14.1. The genome was annotated using the NCBI Prokaryotic Genomes Automatic Annotation Pipeline (PGAAP), and Bacterial Annotation System (BASys) webserver. The proteome of KR2-7 was subjected to BLASTP against the Cluster of Orthologous Group (COGs) database at E-value < 1 × 10^−5^ to identify the Cluster of Orthologous Groups (COGs) [19].

### 2.5. Genome Phylogeny

In this study, 32 *Bacillus* strains belonging to various species were selected among those recorded in the NCBI GenBank database. For all the selected strains, the nucleotide and the corresponding amino acid sequences were retrieved from the GenBank database. Whole-genome alignments were performed using REALPHY (http://realphy.unibas.ch (accessed on 22 December 2021); [20]) and the phylogenetic tree was constructed using the MEGA v. 7 [21] by maximum likelihood method [22], with evolutionary distances computed using the general time-reversible model [23]. Branch validity was evaluated by the bootstrap test with 1000 replications. The average nucleotide identity (ANI) values of selected *Bacillus* strains were calculated using the server EzBioCloud (http://www.ezbiocloud.net/tools/ani (accessed on 21 August 2021); [24]). According to the algorithm developed by Goris et al. [25], 95∼96% cut-off value was used for the species boundary [26]. The web-based DSMZ service (http://ggdc.dsmz.de (accessed on 7 January 2022); [27]) with 70% species and sub-species cut-off was used to estimate the in silico genome-to-genome distance values for the selected strains.

### 2.6. Pathway Analysis

The annotated genome was analyzed using KEGG (Kyoto Encyclopedia of Genes and Genomes) to determine the existing pathways, which were then manually validated through matching the assigned gene functions to the corresponding KEGG pathway.

### 2.7. Genome-Wide Identification of Secondary Metabolite Biosynthesis Gene Clusters

The antibiotics and secondary metabolite analysis shell (antiSMASH) is a comprehensive resource that allows the automatic genome-wide identification and analysis of secondary metabolite biosynthesis gene clusters in bacterial and fungal genomes [28,29]. Thereby, the KR2-7 genome was submitted to the antiSMASH web server (https://antismash.secondarymetabolites.org) (accessed on 22 December 2021) to detect the putative BGCs for secondary metabolites. Each identified BGC in the KR2-7 genome was aligned against corresponding BGC in *B. subtilis* strain 168 and *B. amyloliquefacience* stain FZB42 using Geneious Prime v.2021.2.2. to find out the BGCs similarity between KR2-7, 168 and, FZB42.

## 3. Results

### 3.1. General Genomic Features of Strain KR2-7

The assembled genome of *B. inaquosorum* KR2-7 contained 4 contigs, with an *N50* of 2,144,057 bp and 700X sequence coverage. The KR2-7 genome was obtained with a length of 4,248,657 bp, the G+C content of 43.1%, and 4265 predicted genes consisting of 4017 protein-coding genes, 50 rRNA genes, and 83 tRNA genes. Interestingly, strain KR2-7 possesses the larger number of genes contributing to amino acid transport and metabolism (322 genes), carbohydrate transport and metabolism (278 genes), inorganic ion transport and metabolism (200 genes), and secondary metabolites biosynthesis, transport and catabolism (76) compared to the reputable biocontrol agent *B. velezensis strain FZB42* (Appendix A). Therefore, the genome content of KR2-7 indicates that the strain has considerable potential as a biocontrol agent. The genome sequence of *B. inaquosorum* KR2-7 was deposited in NCBI GenBank under the accession number QZDE00000000.2.

### 3.2. Genome Phylogeny

The genomes of 31 *Bacillus* strains were selected for aligning with the KR2-7 genome and phylogenomic analysis. The selected strains and their corresponding genome sequence accession numbers were presented in Table 1. Selected *Bacillus* strains were accurately distributed on branches of the maximum likelihood phylogenomic tree (Figure 1).

Moreover, closely related *Bacillus* species such as *B. amyloliquefaciens* and *B. velezensis* were distributed on the same branch (Figure 1). The genome-based phylogeny approaches well recognized *B. methylotrophicus*, *B. amyloliquefaciens* subsp. *plantarum*, and *B. oryzicola* as the heterotypic synonyms of *B. velezensis* [30]. Recently, three subspecies of *Bacillus subtilis*, including *B. subtilis* subsp. *inaquosorum*, *B. subtilis* subsp. *Spizizenii*, and *B. subtilis* subsp. *stercoris* were promoted to species status through comparative genomics. Each subspecies encompasses unique bioactive secondary metabolite genes which cause the unique phenotypes [31]. According to REALPHY results, strain KR2-7 was identified as *B. inaquosorum*, owing to being placed within the *B. subtilis* branch close to *B. subtilis* subsp. *inaquosorum* strain KCTC 13429 and strain DE111 in the phylogenomic tree (Figure 1). Notably, the results of ANI and GGDC analysis were consistent with REALPHY results as the KR2-7 genome displayed the highest ANI values (99.26%) and the lowest GGDC values (0.0075) with respect to the genome of strain KCTC 13429 (Table 1). Interestingly, the phylogeny analysis of several *B. amyloliquefaciens* strains based on core-genome was consistent with the ANI and GGDC values [32]. Altogether, the aforesaid genome-based phylogeny approaches identified strain KR2-7 as *Bacillus inaquosorum*.

### 3.3. Secondary Metabolites Biosynthetic Gene Clusters in the KR2-7 Genome

Genome mining of the strain KR2-7 revealed that more than 700 kb (i.e., nearly 17% of the genome) is devoted to 13 putative BGCs. Of the 13 found BGCs, nine were identified to contain one polyketide synthase (PKS) for macrolactin; five non-ribosomal peptide synthetases (NRPSs) for bacillibactin, bacillomycin F, bacilysin, fengycin, and surfactin; one PKS-NRPS hybrid synthetases (PKS-NRPS) for bacillaene; one thiopeptide synthase for subtilosin A, and one head-to-tail cyclised peptide for the sporulation killing factor. The nine annotated BGCs encode secondary metabolites which contribute to plant growth promotion through the fungal/bacterial pathogen suppression, ISR, nutrient uptake, and plant colonization (Table 2) [33,34,35,36]. The distribution of identified BGCs within the KR2-7 genome underlies its vigorous potential in plant disease biocontrol application [16]. The coding genes of secondary metabolites in KR2-7 were different from those in *B. velezensis* FZB42, while these genes showed more similarity with those in *B. subtilis* 168 (Table 2). Interestingly, the BGC of bacillomycin F in KR2-7 was absent in *B. subtilis* 168 and *B. velezensis* FZB42 (Table 2) as this gene cluster conserved in *B. inaquosorum* [31]. Moreover, the KR2-7 genome contains four unannotated BGCs (data not shown) which showed less similarity to compounds listed in the MIBiG database.

### 3.4. Antifungal Secondary Metabolites Production in Strain KR2-7

The KR2-7 genome mining showed that this strain harbors three BGCs with antifungal function, including fengycin, surfacing, and bacillomycin F (a variant of iturin) belonging to *Bacillus* cyclic-lipopeptides (CLPs). *Bacillus* CLPs represented the powerful fungitoxicity properties by interfering with cell membrane integrity, permeabilizing the cell membrane, and perturbing membrane osmotic balance due to the formation of ion-conducting pores [37].

Strain KR2-7 not only suppressed the Fusarium wilt of tomato caused by *Fusarium oxysporum* f. sp. *lycopersici* [16] but also showed a broad-spectrum antifungal activity towards various phytopathogenic fungi including *Alternaria alternata* f. sp. *lycopersici*, *Athelia rolfsii*, *Botrytis cinerea*, *Rhizoctonia solani*, and *Verticillium albo-atrum* (Figure 2).

Fengycin (plipastatin), the powerful fungitoxic compound—especially against filamentous fungi [37]—is synthesized by NRPS and encoded by a 39302 bp gene cluster with five genes including *pps*A-E in KR2-7, which showed a 92% and 72.05% similarity to the fengycin gene cluster of *B. subtilis* 168 and *B*. *amyloliquefacience* FZB42, respectively (Table 2). The first three genes (*pps*ABC) each encode two amino acid modules. The fourth gene (*pps*D) encodes three amino acid modules, and the last gene (*pps*E) encodes one amino acid module (Figure 3). Ions of m/z values 1471.8, 1485.7, 1487.9, 1499.9, 1501.9, 1513.8, 1515.9, 1527.8 1529.9 and, 1543.8 were observed in a whole-cell surface extract of KR2-7 grown on a dual culture plate (thereafter, dual culture cell extract) and assigned to C15 to C18 fengycin homologues while in the whole-cell surface extract of KR2-7 grown on the control plate (thereafter, control cell extract) only four aforesaid peaks (m/z 1501.9, 1515.9, 1529.9 and, 1543.8) were detected (Table 3, Appendix A). The result indicated that the KR2-7 strain secreted various fengycin homologues to inhibit the growth of *Fol*.

More strikingly, a 37074 bp gene cluster encoding bacillomycin F was also identified immediately downstream of the fengycin gene cluster of KR2-7 (Figure 4). The bacillomycin F is one of seven main variants within the iturin family [40] encoded by a gene cluster consisting of four genes designated *itu*D, *itu*A, *itu*B and, *itu*C. The gene cluster code a cyclic heptapeptide in which the first three amino acids are shared among iturin family members, whereas the remaining four amino acids are conserved in *B. inaquosorum* [31]. Furthermore, iturins are characterized by a heptapeptide of α-amino acids attached to a β-amino fatty acid chain with a length of 14 to 17 carbons [37]. They possess potent antifungal activity against a wide variety of fungi and yeast, but bounded antibacterial and no antiviral actions [41,42,43]. Furthermore, these molecules also showed strong haemolytic activity, which limits their clinical use [44]. The antifungal mechanism of iturins launches by their interaction with the target cell membrane and osmotic perturbation of the membrane, owing to the formation of ion-conducting pores. Subsequently, the change in the permeability of a membrane is conducive to the release of biomolecules, such as proteins, nucleotides, and lipids from cells, which ultimately causes cell death [44,45]. In the dual culture cell extract of KR2-7, six mass peaks assigned to C16, C18 and C19 forms of iturin were observed while they were absent in the control cell extract of KR2-7 (Table 4, Appendix A). This result indicated that strain KR2-7 produced different variants of iturins to limit the growth of *Fol* hyphae.

Similar to fengycin, surfactin was synthesized by NRPS and encoded by a *srf* gene cluster that spans 26073 bp in the KR2-7 genome. The gene cluster harbors four genes (*srf*AA-AD) and showed a 92.20% and 74.65% similarity to those of *B. subtilis* 168 and *B*. *amyloliquefacience* FZB42, respectively. The product of the *srf* gene cluster is a linear array of seven modules, six of which are encoded by *srf*AA and *srf*AB genes and the last module is encoded by *a srf*AC gene (Figure 5). The fourth gene (*srf*AD) encodes thioesterase/acyltransferase (Te/At-domain) which triggers surfactin biosynthesis [37]. Hence, the *sfp* gene encodes an essential enzyme (phosphopantetheinyl transferase) for the non-ribosomal synthesis of lipopeptides and the synthesis of polyketides. The regulatory gene *ycz*E encoding an integral membrane protein was detected within the KR2-7 genome (Figure 5). Surfactin enables bacteria cells to interact with plant cells as a bacterial elicitor for stimulating ISR [37], especially through the activation of jasmonate- and salicylic acid-dependent signaling pathways [46]. Several studies indicated the ISR-elicitor role of surfactin against phytopathogens in various host plants, e.g., tomato [47], wheat [48], citrus fruit [49], lettuce [50], and grapevine [51]. Comparing the MALDI-TOF mass spectra of KR2-7 grown on a PDA control and dual culture revealed that surfactin contributed to the suppression of *Fol* as eight mass peaks assigned to C13, C14 and C15 surfactin homologs were detected in dual culture cell extracts, while only four of which were observed in the control cell extract (Table 5, Appendix A). Furthermore, the bacterium produced more surfactin to suppress *Fol*. Additionally, C14 and C15 surfactin tend to stimulate stronger ISR rather than those with shorter chain lengths [47]. Moreover, suppression of taxonomically diverse fungal pathogens including *Fusarium oxysporum*, *F. moniliforme*, *F. solani*, *F. verticillioides*, *Magnaporthe grisea*, *Saccharicola bicolor*, *Cochliobolus hawaiiensis*, and *Alternaria alternata* by the surfactin family demonstrated that surfactins are strong fungitoxic compounds [52,53,54,55].

### 3.5. Antibacterial Secondary Metabolites Production in Strain KR2-7

The KR2-7 genome contained six BGCs coding for antibacterial compounds including bacillaene and macrolactin, sporulation killing factor (skf), subtilosin A, bacilysin, and surfactin. Several studies on surfactin and its isoforms proved that these metabolites played a major role in combating bacterial plant diseases, such as fruit bloch caused by *Acidovorax citrulli* in melon [56], tomato wilt caused by *Ralstonia solanacearum* [57], and root infection by *Pseudomonas syringae* in *Arabidopsis* [58]. Moreover, surfactin produced by *B. subtilis* R14 exhibited pronounced antagonistic efficacy against several multidrug-resistance bacterial strains of *Escherichia coli*, *Pseudomonas aeruginosa*, *Staphylococcus aureus*, and *Enteroccoccus faecalis* [59]. 

Bacillaene is a polyketide known as a selective bacteriostatic agent that inhibits prokaryotic, not eukaryotic growth by disrupting protein synthesis [60]. Its antimicrobial efficacy against various bacteria (*Myxococcus xanthus* and *Staphylococcus aureus*) and fungi (*Fusarium* spp) have been reported [60,61,62]. In the KR2-7 genome, bacillaene was synthesized by a PKS/NRPS hybrid pathway and encoded by a giant *pks* gene cluster (76.355 Kbp) containing 16 genes (*pks*A-S and *acp*K) showing an 89.63% and 75.25% similarity to those of *B. subtlis* 168 and *B. velezensis* FZB42, respectively (Table 2, Figure 6B). Another polyketide, macrolactin, can be encoded by a 54.225 kbp gene cluster in strain KR2-7 and showed a 74.12% similarity to *the mln* cluster of *B. velezensis* FZB42 (Table 2, Figure 6A). Macrolactins are a large class of macrolide antibiotics that inhibited the growth of several bacteria, including *Ralstonia solanacearum*, *Staphylococcus aureus*, and *Burkholderia epacian* [63,64]. In the dual culture cell extract of KR2-7, one ion corresponding to 7-o-succinyl macrolactin A ([M + Na]^+^ = 525.4), and another ion corresponding to bacillaene A ([M + H]^+^ = 583.5) were detected (Figure 6C,D), while they were not observed in the control cell extract of KR2-7.

Bacilysin (also known as tetaine) is a dipeptide suppressing a wide variety of destructive phytopathogenic bacteria, e.g., *Erwinia amylovora*, *Xanthomonas oryzae* pv. *oryzae*, *X. oryzae* pv. *Oryzicola*, and *Clavibacter michiganense* subsp. *sepedonicum* [65,66,67]. This bactericidal property is due to the inhibition of glucosamine-6- phosphate synthase by the anticapsin moiety of bacilysin. Its inhibition represses the biosynthesis of peptidoglycans, the essential constituents of the bacterial cell wall [68,69]. In the KR2-7 genome, bacilysin can be encoded by a 7128 bp *bac* gene cluster consisting of seven genes (*bac*A-E, *ywf*AG), and display high gene similarity to those of *B. subtilis* 168 (Table 2, Figure 7). This metabolite and its derivatives were detected neither in the KR2-7 control cell extract nor dual culture cell extract, likely due to culture conditions or the assay method.

Furthermore, the KR2-7 genome encompassed two distinct gene clusters encoding bacteriocins, including subtilosin A and sporulation killing factors (SKFs). Subtilosin A is a macrocyclic anionic antimicrobial peptide originally obtained from wild-type strain *B. subtilis* 168 [70] but is also produced by *B. amyloliquefaciens* and *B. atrophaeus* [71,72]. This bacteriocin displayed a bactericidal effect on a broad spectrum of bacteria, including Gram-positive and Gram-negative bacteria and both aerobes and anaerobes [73], possibly through an interaction with membrane-associated receptors, or binding to the outer cell membrane, and is conducive to membrane permeabilization [73,74,75]. Subtilosin A is ribosomally synthesized by an *alb* gene cluster containing eight genes (*alb*A-G, *sbo*A) spanning 6.8 kbp in the KR2-7 genome (Figure 8). The *sbo*A gene encodes presubtilosin, and *alb*A-G genes encode proteins whose functions are presubtilosin processing and subtilosin export [76]. The mass peaks corresponding to subtilosin A and its homologs appeared neither in the KR2-7control cell extract nor the dual culture cell extract. These peaks are detectable by altering the culture condition and/or evaluating the procedure.

The KR2-7 genome also harbored a 5976 bp *skf* gene cluster encompassing *skf*ABCEFGH, and involves the production and release of killing factors during sporulation (Figure 9). During the early stages of sporulation, sporulating cells of *B. subtilis* exude extracellular killing factors to kill the nonsporulating sister cells whose immunity to these toxins was not developed. As a result, the nutrient from the dead cells are released and then used by the sporulating cells to resume their growth. This phenomenon is termed “cannibalism” and causes a delay in sporulation [77,78]. The SKF bacteriocin produced by the sporulating cells can destroy other soil-inhabiting bacteria. Similarly, the expression of *skf* genes in *B. subtilis* inhibits the growth of *X. orzae* pv. *oryzae*, the causative agent of rice bacterial blight [79].

### 3.6. Plant Colonization by Strain KR2-7

The most crucial step for a PGPR (Plant Growth Promoting Rhizobacteria) agent to survive, enhance plant growth, and suppress plant disease is the efficient colonization of plant tissues. The plant colonization process comprises two steps. In the first step, PGPR agents reach the surface of plant tissue either by passive movement in water flow or by flagellar movement. The second step is to establish the plant-bacterium interaction reliant on bacterial biofilm formation [36,80].

The KR2-7 genome harbored the gene clusters for flagellar assembly (*flg* cluster, *flh* cluster and, *fli* cluster) and bacterial chemotaxis (*che* cluster) together with other genes known to be necessary for swarming motility, including *hag*, two stator elements (*mot*AB), as well as regulatory genes *swr*AA, *swr*AB, *swr*B and, *swr*C (Table 6). In the step of efficient colonization, the PGPR agent forms bacterial biofilm and not only strengthens the plant-bacterium interaction but protects the plant root system as a bio-barrier against pathogen attacks [80]. The main component of bacterial biofilm is the extracellular polymeric substances (EPS) with a chemical composition including proteins, neutral polysaccharides, charged polymers, and amphiphilic molecules [80]. The *eps* cluster (*eps*C-O) encoding exopolysaccharide of biofilm and its regulatory genes *sin*R and *arb*A (repressors) and, *sin*I (antirepressor), the *yqx*M-*sip*W-*tas*A gene cluster encoding amyloid fiber (TasA protein of biofilm) and *pgc*A encoding phosphoglucomutase were found in the KR2-7 genome (Table 6). Moreover, the involvement of surfactin in cell adhesion and biofilm formation due to its 3D topology and amphiphilic nature has been illustrated [81,82]. Baise et al. [58] declared that deleting surfactin gene expression in *B. subtilis* strain 6051 led to disability to form robust biofilm on *Arabidopsis* root surface, and reduced the suppression of disease caused by *Pseudomonas syringae.* Besides, the deficiency in surfactin production in *B. subtilis*, strain UMAF6614 resulted in a biofilm formation defect on melon phylloplane and partially reduced the suppression of bacterial soft root rot, bacterial leaf spot, and cucurbit powdery-mildew by the biocontrol stain [83].

### 3.7. Genes Involved in Bacterium-Plant Interactions

Quite apart from the antagonistic mechanisms of bacterial biocontrol strains, these bacterial microorganisms are also involved in plant growth augmentation through making nutrients available for host plants, production of plant growth-promotion hormones, and the induction of systemic resistance within the plant by specific metabolite secretion [96,97]. Similar to other biocontrol microorganisms, the KR2-7 genome contains the genes/gene clusters related to plant growth promotion (Table 6).

The KR2-7 genome contained *moa*A-E genes encoding molybdenum cofactor and may be a relic of a nitrogen-fixing gene cluster or a cofactor for nitrogen assimilation [80]. Moreover, the genes for nitrate reduction (*nar*G-J), nitrate transport (*nar*K), probable transcription regulator genes (*arf*M), regulatory protein (*fnr*), an ammonium acid transporter (*nrg*A), and its regulator gene (*nrg*B), along with the *nas* gene cluster (*nas*A-F), were also identified in the KR2-7 genome. The *nas* gene cluster is involved in nitrite transport and reduction (Table 6). 

In addition to nitrogen assimilation, the KR2-7 genome encompassed potassium transporting genes, including *ktr* system potassium uptake proteins (*ktr*A-D), a putative potassium channel protein (*yug*O), and putative gamma-glutamyl cyclotransferase (*ykq*A) [80,98]. Furthermore, the presence of genes for transportation of magnesium (*mgt*E, *cor*A), ferrochrome (*yvd*K), manganese (*mnt*H), and a gene cluster for manganese binding/transport (*mnt*A-D), along with the transcription regulator protein (*mnt*R), were identified in the KR2-7 genome. These genes uptake the nutrients or detoxify the heavy metal ions for both the bacteria and host plants [80]. An 11.7 kb *dhb* gene cluster (*dhb*ABDEF) encoding siderophore bacillibactin was identified in the KR2-7 genome.

Furthermore, ions of m/z values 883.4 and 905.2 were detected in the KR2-7dual culture cell extract and were identified as bacillibactin [M + H]^+^ and bacillibactin [M + Na]^+^ (Figure 10) by comparison with previously reported data [99,100]. Notably, the molecular ion peaks corresponding to bacillibactin were not observed in the control cell extract of KR2-7. Siderophores are low-molecular-weight molecules with a high affinity for ferric iron that solubilize iron from minerals and organic compounds under iron limitation conditions [101]. Siderophore-producing bacterial strains impact plant health by supplying iron to the host plants [102,103], depriving fungal pathogens of iron [104], and suppressing fungal phytopathogens, including *F. oxysporum* f. sp. *capsici* [105] and *Phytophthora capsici* [106]. Furthermore, it was reported that siderophores mitigate heavy metal contamination of soil through the formation of a stable complex with environmental toxic metals such as Al, Cd, Cu, Ga, In, Pb, and Zn [101].

Volatile organic compounds (VOCs) produced by PGPR agents play a significant role in promoting plant growth through the regulation of synthesis or metabolism of phytohormones [107], the induction of systemic disease resistance [108,109], and the control of plant pathogens [110]. A 2,3-butanediol and 3-hydroxy-2-butanone (acetoin) are the best-known growth-promoting VOCs that produced *B. subtilis* and *B. amyloliquefaciens*. The genome of the KR2-7 harbored *als* gene cluster (*als*R, *als*S, *als*D), along with the *bdh*A gene is together required for the biosynthesis pathway of 2,3-butanediol from pyruvate. In this pathway, *als*S encodes the acetolactate synthase enzyme, which catalyzes the condensation of two pyruvate molecules into acetolactate. Then, acetolactate decarboxylase, encoded by *als*D, converts decarboxylate acetolactate into acetoin. The *als*R regulates two aforesaid steps. Finally, the *bdh*A encoded (R, R)-butanediol dehydrogenase enzyme catalyzes 3-hydroxy-2-butanone (acetoin) to 2,3-butanediol [111]. In addition, the KR2-7 genome contained *ilv*H, *ilv*B, *ilv*C genes and a *leu* gene cluster (*leu*ABCD) which are required for the biosynthesis pathway of three branched-chain amino acids (BCAA), including leucine, isoleucine, and valine. Acetolactate is a central metabolite between 2,3-butanediol and BCAA biosynthesis and can involve in both anabolism and catabolism by acetolactate decarboxylase. It was reported that acetolactate decarboxylase is an enzyme with a dual role that can direct acetolactate flux to catabolism in favour of valine and leucine biosynthesis or can catalyze the second step of the 2, 3-butanediol anabolism pathway [112].

*Bacillus* spp. can enhance plant growth through the synthesis of plant growth-promoting hormones, such as auxin, indole-3-acetic acid (IAA), and gibberellic acid. The genome of KR2-7 may encompass genes/gene clusters responsible for the biosynthesis of indole acetic acid, phytase, and trehalose (Table 6). Moreover, a large variety of PGPRs produce polyamines, such as putrescine, spermine, spermidine, and cadaverine, and are known to be involved in promoting plant growth and improving abiotic stress tolerance in plants [113]. The genes coding for arginine decarboxylase (*Spe*A), agmatinase (*Spe*B), and spermidine synthase (*Spe*E), which direct polyamines biosynthesis, were also found in the KR2-7 genome (Table 6).

## 4. Discussion

Previously, the *Bacillus subtilis* species complex was composed of four close subspecies, i.e., subspecies *subtilis*, *spizizenii*, *inaquosorum*, and *stercoris*, which were differentiated through a phylogenetic analysis of multiple protein-coding genes and genome-based comparative analysis [114,115]. *B. subtilis* subsp. *Inaquosorum* was deemed as a distinctive taxon encompassing strains KCTC 13429 and NRRL B-14697 [116]. Recent phylogenomic studies clearly distinguished subspecies *inaquosorum* from subspecies *spizizenii*, as the estimated ANI among them was smaller than the defined ANI for species delineation (95% ANI) [117]. In addition to a low ANI value (<95%), the BGC of subtilin exclusively presents in the genomes of subspecies *spizizenii*, but was not characterized in subspecies *inaquosorum* genome [115]. Accordingly, *B. inaquosorum* KR2-7 was clearly differentiated from *B. subtilis* subsp. *spizizenii* W23 because of the low ANI value among them (94.18%) and the lack of subtilin gene cluster in the genome content of strain KR2-7. In addition, it was reported that *B. inaquosorum* is the only species to produce bacillomycin F. It was approved by detecting a unique MALDI-TOF-MS biomarker at m/z 1120 in the MALDI-TOF-MS spectra of *B. inaquosorum* that cannot be produced by other species [114]. Since this unique biomarker (m/z value 1120.6) was observed in the MALDI-TOF-MS spectra of strain KR2-7, it can be concluded that this strain is a *B. inaquosorum.* Recently, the ability of *B. inaquosorum* strain HU Biol-II in producing bacillomycin F was confirmed through HPLC MS/MS [15]. Most recently, subspecies *spizizenii*, *inaquosorum*, and *stercoris* were promoted to species status through a comparative genome study [118]. This study determined that each subspecies had unique secondary metabolite genes encoding unique phenotypes, thereby each subspecies can be promoted to species. According to the aforesaid results, strain KR2-7 was identified as a *B. inaquosorum*.

The genome-driven data highlighted the plant-beneficial functions of strain KR2-7. This strain can efficiently colonize the plant root surface, relying on its swarming motility and biofilm formation abilities. Efficient root colonization of biocontrol bacteria is necessary for suppressing phytopathogens, and biofilm formation is an essential prerequisite for persistent root colonization [119,120]. The biofilm-deficient mutant of *B. pumilus* HR10 produced weakened biofilms with reduced contents of extracellular polysaccharides and proteins, and thereby could not efficiently control pine seedling damping-off disease [121]. Hence, the suppression of tomato Fusarium wilt by strain KR2-7 [16] may contribute to efficient tomato root colonization of this strain.

In addition to efficient root colonization, strain KR2-7 is able to directly suppress soil-dwelling phytopathogens through producing eight antimicrobial secondary metabolites, e.g., fengycin, surfactin, bacillomycin F, macrolactin, bacillaene, bacilysin, subtilosin A, and sporulation killing factor. The combination of obtained data via MALDI-TOF-MS with our previous observations [16] confirmed that strain KR2-7 produced at least four bioactive metabolites (including fengycin, surfactin, macrolactin, and bacillaene) to directly protect the tomato plant from the invasion and penetration of *Fol.* The cyclic lipodecapeptide fengycin exhibits strong fungitoxic properties by inhibiting phospholipase A2 and aromatase functions [122], disruption of biological membrane integrity [123], deformation and permeabilization of hyphae [124,125], and induction of ISR [126]. In this context, the strong antifungal activity of *B. inaquosorum* strain HU Biol-II against a diverse group of fungi highly pertained to the fengycin produced by this strain. Interestingly, 97.47% of *the pps*A-E gene cluster in KR2-7 was similar to the fengycin gene cluster in strain HU Biol-II [15]. Fengycin produced by *B. subtilis* SQR9 and *B. amyloliquefaciens* NJN-6 significantly inhibited the growth of *F. oxysporum* [127,128]. Moreover, fengycin BS155 isolated from *B. subtilis* BS155 destroyed *Magnaporthe grisea* through damaging the plasma membrane and cell wall, disruption of mitochondrial membrane potential (MMP), chromatin condensation, and the induction of reactive oxygen species (ROS) [129]. In addition to fengycin, the contribution of other secondary metabolites in the biocontrol of various pathogens has been reported. The supernatant of *B. subtilis* GLB191, consisting of surfactin and fengycin, highly controlled grapevine downy mildew caused by *Plasmopara viticola* by means of direct antagonistic activity and the stimulation of plant defence [51]. Furthermore, the strong antifungal effect of *B. velezensis* strains Y6 and F7 against *Ralstonia solanacearum* and *F. oxysporum* was attributed to the production of fengycin, iturin, and surfactin, among which iturin played a key role in the suppression of *F. oxysporum* [130]. The biocontrol mechanism of *B. amyloliquefaciens* DH-4 against *Penicillium digitatum*, the causal agent of citrus green mold, was secreting a cocktail of antimicrobial compounds consisting of macrolactin, bacillaene, iturins, fengycin, and surfactin [100].

Additionally, strain KR2-7 can exert hormones, such as IAA, phytase, and trehalose for root uptake and rebalance hormones in the host plant to boost growth and stress response. Phytate (inositol hexa- and penta-phosphates) is the predominant form of soil organic phosphorus, which is unavailable for plant uptake due to the rapid immobilization of phosphorus and the lack of adequate phytase levels in plants [131]. Phytase is a phosphatase enzyme responsible for the transformation of organic phytate to inorganic phosphate, which is acquirable for plant roots. Similarly, phytase-producing Bacillus strains can effectively enhance plant growth through the liberation of reactive phosphorus from phytate and make this element available for plant uptake. In the presence of phytate, the comparison of the culture filtrate of B. amyloliquefaciens strain FZB45 with those of a phytase-deficient mutant provided evidence that the phytase activity of strain FZB45 enhanced the growth of corn seedling [132]. The bacterization of Brassica juncea with Bacillus sp. PB-13 considerably boost phosphorus content and growth parameters in 30-day-old seedlings [133]. More recently, the soil inoculation of Bacillus strain SD01N-014 resulted in the enhancement of soil phosphorus content and the promotion of maize seedling growth [134]. Accordingly, extracellular phytase activity of strain KR2-7 mediated with phy gene can be expectable. In addition, the presence of genes involved in the biosynthesis of IAA and trehalose in the KR2-7 genome (Table 6) is an indication of this strain’s potential in the mitigation of salt toxic stress on plants. Inoculation of tomato plants subjected to salt stress with OxtreS (trehalose over-expressing strain) mutant of Pseudomonas sp. UW4 considerably boosted the dry weight, root and shoot length, and chlorophyll content of the tomato plant [135]. Moreover, canola seedlings treated with over-expressed IAA transformant of UW4 represented longer primary root with an increased number of root hairs than seedlings treated with wild-type UW4 [136]. The growth promotion of root hairs by IAA improves the assimilation of water and nutrients from the soil, which in turn raises plant biomass [136]. Similarly, Japanese cypress seedlings inoculated with B. velezensis CE 100 showed significant increases in growth parameters and biomass due to the production of indole-3-acetic acid (IAA) by CE 100 strain [137].

## 5. Conclusions

According to genome-driven data, along with chemical analysis results, strain KR2-7 most likely exploits four possible modes of action to control tomato Fusarium wilt, as shown in Figure 11:(1)Inhibition of the pathogen growth through the diffusion of antifungal and antibacterial secondary metabolites and biofilm formation;(2)Stimulation of ISR in tomato via the production of surfactin and volatile organic compounds;(3)Promotion of plant health and growth by producing plant growth promotion hormones and polyamines, supplying iron for tomato, depriving the pathogen of iron, and relieving heavy metal stress in the soil as a result of siderophore bacillibactin activity;(4)Efficient colonization of plant roots.

The described modes of action were highly based on the identified gene clusters encoding secondary metabolites and characterized gene/gene clusters involved in plant colonization, plant growth promotion, and ISR. Furthermore, future studies using integrated omics approaches and the mutagenesis of strain KR2-7 are required to approve the aforesaid possible modes of action of strain KR2-7 and exact functions of the putative genes and gene clusters in the suppression of fungal pathogen *Fol*.

## Figures and Tables

**Figure 1 biology-11-00137-f001:**
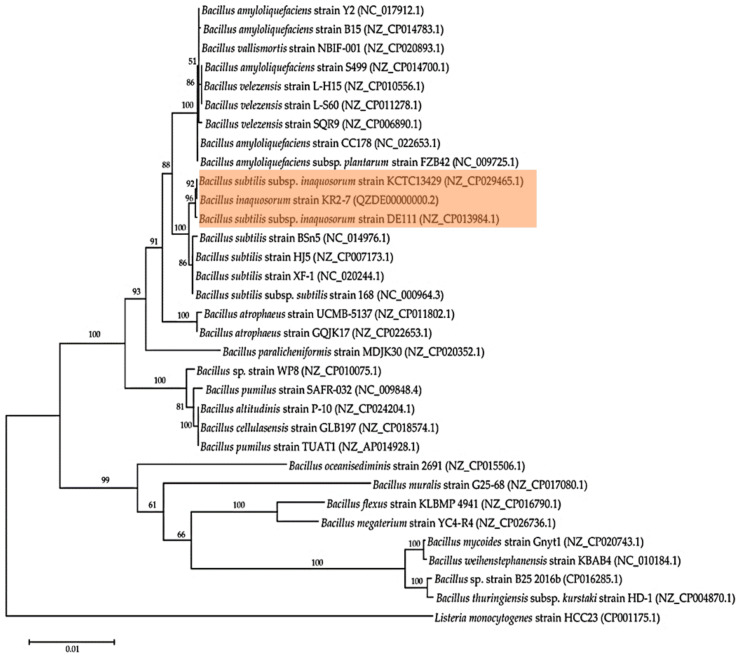
Maximum Likelihood phylogenomic tree of strain KR2-7 and selected *Bacillus* strains based on REALPY. Numbers at nodes represent the percentages of occurrence of nodes in 1000 bootstrap trials. The *Listeria monocytogenes* strain HCC23 (CP001175.1) was served as an outgroup.

**Figure 2 biology-11-00137-f002:**
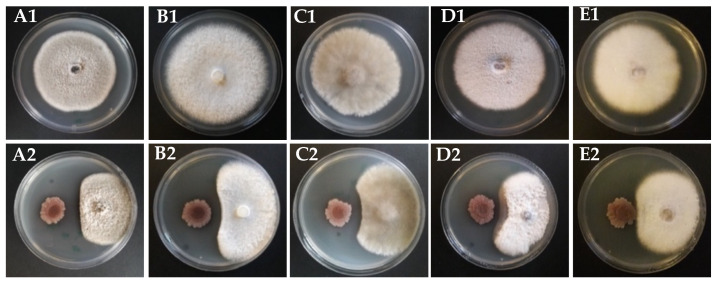
Antifungal activity of strain KR2-7 towards various phytopathogenic fungi. (**A1**–**E1**): a 5-mm agar plug of each phytopathogenic fungi including *Alternaria alternata*, *Athelia roflsii*, *Botrytis cinerea*, *Rhizoctonia solani*, and *Verticillium albo-atrum* was cultured on the center of the PDA plate for 6 days at 28 °C, respectively. (**A2**–**E2**): strain KR2-7 was simultaneously cultured 3cm apart from the plug of (**A2**): *Alternaria alternata*, (**B2**): *Athelia roflsii*, (**C2**): *Botrytis cinerea*, (**D2**): *Rhizoctonia solani*, (**E2**): *Verticillium albo-atrum*.

**Figure 3 biology-11-00137-f003:**
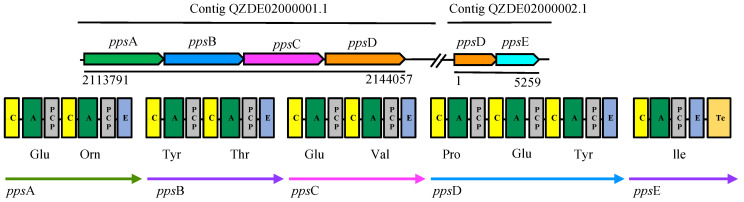
The biosynthetic gene cluster of fengycin in strain KR2-7.

**Figure 4 biology-11-00137-f004:**
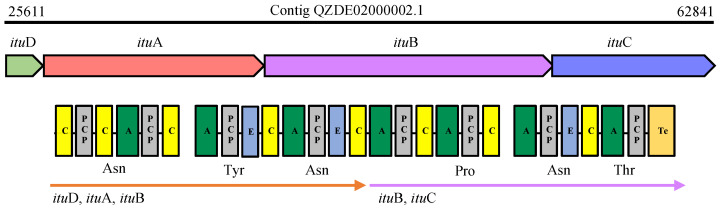
The biosynthetic gene cluster of bacillomycin F in strain KR2-7.

**Figure 5 biology-11-00137-f005:**
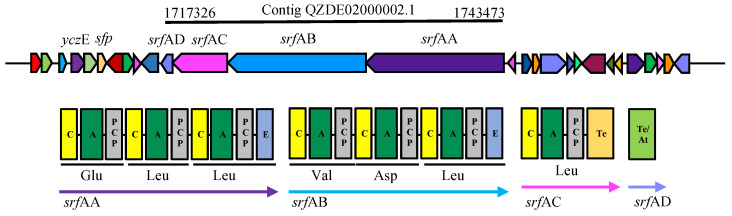
The biosynthetic gene cluster of surfactin in strain KR2-7.

**Figure 6 biology-11-00137-f006:**
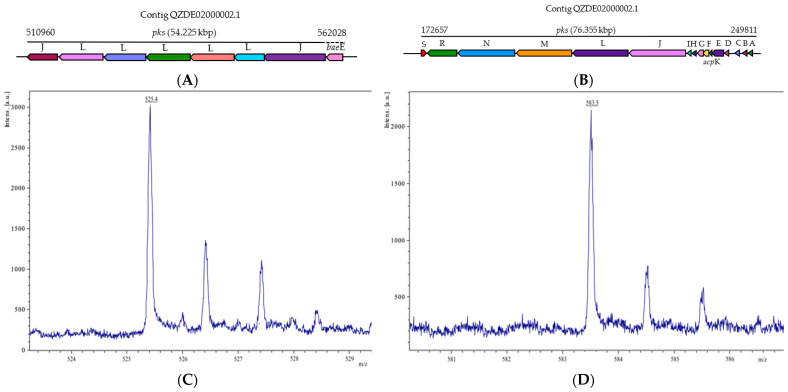
The biosynthetic gene clusters of (**A**) macrolactin and (**B**) bacillaene in strain KR2-7 and MALDI-TOF MS analysis of antibacterial secondary metabolites produced by strain KR2-7 grown on a dual culture plate. (**C**) m/z 525.4: 7-o-succinyl macrolactin A [M + Na]+; (**D**) m/z 583.5: Bacillaene A [M + H]+.

**Figure 7 biology-11-00137-f007:**
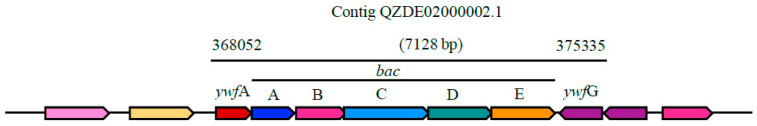
The biosynthetic gene cluster of bacilysin antibacterial metabolite in strain KR2-7.

**Figure 8 biology-11-00137-f008:**
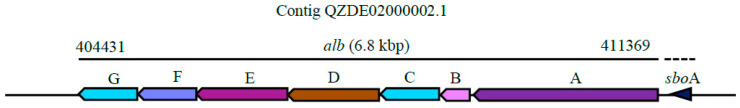
The biosynthetic gene cluster of subtilosin A antibacterial metabolite in strain KR2-7.

**Figure 9 biology-11-00137-f009:**
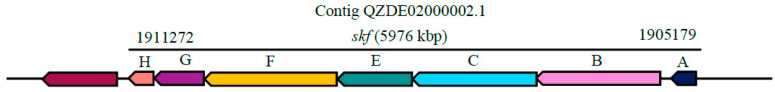
The biosynthetic gene cluster of sporulation killing factor antibacterial metabolite in strain KR2-7.

**Figure 10 biology-11-00137-f010:**
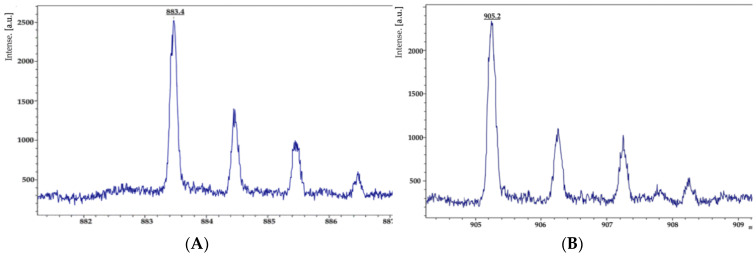
MALDI-TOF MS analysis of bacillibactin produced by strain KR2-7 grown on a PDA dual culture. (**A**) m/z 883.4: bacillibactin [M + H]^+^; (**B**) m/z 905.2: bacillibactin [M + Na]^+^.

**Figure 11 biology-11-00137-f011:**
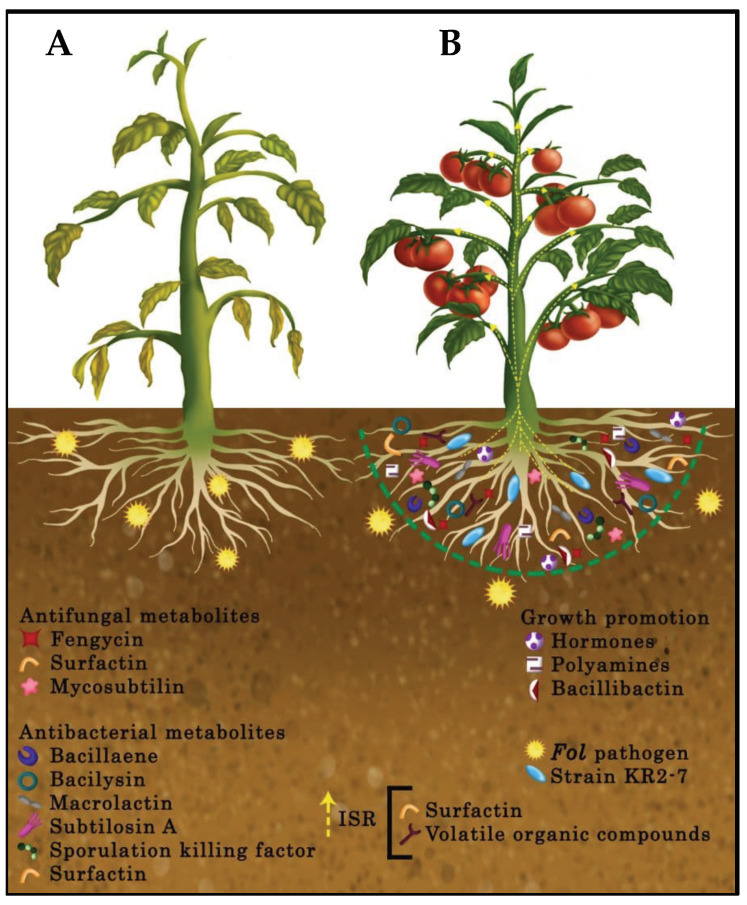
Schematic presentation of putative biocontrol mechanism of strain KR2-7 against Fol. (**A**) An untreated tomato plant in which *Fol* (yellow 16-point star) penetrated root tissue, colonized and blocked the vascular system to prevent water and nutrients from being transferred to plant organs. It caused yellowing began with bottom leaves, followed by wilting, browning, and defoliation. Growth is typically stunted, and little or no fruit develops. (**B**) Strain KR2-7 (blue rod) reaches to tomato root and colonizes on the root surface through its motility potential and biofilm formation. As a result of root colonization, strain KR2-7 diffuses a wide variety of antifungal and antibacterial secondary metabolites to establish a protective zone (green dash line semicircular) in the tomato rhizosphere. Strain KR2-7 directly limits the invasion of *Fol* fungal pathogen through diffused antifungal secondary metabolites and also control the bacterial pathogens of tomato by means of produced antibacterial secondary metabolites. Meanwhile, volatile organic compounds and surfactin stimulate tomato systemic resistance to provide ISR-mediated protection (yellow dash line arrow) against phytopathogens. Moreover, tomato growth is enhanced assisted by growth-promoting hormones, polyamines, and siderophore bacillibactin.

**Table 1 biology-11-00137-t001:** Average nucleotide identity (ANI) and Genome-to-Genome Distance Calculation (GGDC) values between each selected Bacillus strain and strain KR2-7.

Species	Strain	Accession Number	ANI (%)	GGDC
*Bacillus altitudinis*	P-10	NZ_CP024204.1	71.39	0.2368
*Bacillus amyloliquefaciens*	B15	NZ_CP014783.1	77.31	0.2087
	CC178	NC_022653.1	77.35	0.2082
	FZB42	NC_009725.1	77.65	0.2081
	L-H15	NZ_CP010556.1	77.38	0.2104
	L-S60	NZ_CP011278.1	77.37	0.2101
	S499	NZ_CP014700.1	77.36	0.2097
	Y2	NC_017912.1	77.43	0.2086
*Bacillus atrophaeus*	GQJK17	NZ_CP022653.1	80.4	0.1895
	UCMB-5137	NZ_CP011802.1	80.3	0.192
*Bacillus cellulasensis*	GLB197	NZ_CP018574.1	71.45	0.2368
*Bacillus flexus*	KLBMP 4941	NZ_CP016790.1	68.97	0.1585
*Bacillus megaterium*	YC4-R4	NZ_CP026736.1	68.87	0.1619
*Bacillus muralis*	G25-68	NZ_CP017080.1	68.82	0.1484
*Bacillus mycoides*	Gnyt1	NZ_CP020743.1	68.46	0.1317
*Bacillus oceanisediminis*	2691	NZ_CP015506.1	69.26	0.1506
*Bacillus paralicheniformis*	MDJK30	NZ_CP020352.1	73.09	0.2242
*Bacillus pumilus*	SAFR-032	NC_009848.4	71.3	0.2341
*Bacillus pumilus*	TUAT1	NZ_AP014928.1	71.2	0.2366
*Bacillus* sp.	B25 (2016b)	CP016285.1	68.41	0.177
	WP8	NZ_CP010075.1	71.11	0.2347
*Bacillus subtilis*	BSn5	NC_014976.1	93.06	0.0706
	HJ5	NZ_CP007173.1	93.1	0.0703
	XF-1	NC_020244.1	93.01	0.0707
*Bacillus subtilis* subsp. *inaquosorum*	KCTC 13429	NZ_CP029465.1	99.26	0.0075
*Bacillus subtilis* subsp. *inaquosorum*	DE111	NZ_CP013984.1	98.8	0.01222
*Bacillus subtilis* subsp. *spizizenii*	W23	NC_014479.1	94.18	0.0585
*Bacillus subtilis* subsp. *subtilis*	168	NC_000964.3	93.03	0.0707
*Bacillus thuringiensis* subsp. *kurstaki*	HD-1	NZ_CP004870.1	68.83	0.1575
*Bacillus vallismortis*	NBIF-001	NZ_CP020893.1	77.57	0.2081
*Bacillus velezensis*	SQR9	NZ_CP006890.1	77.38	0.2095
*Bacillus weihenstephanensis*	KBAB4	NC_010184.1	68.46	0.1386

**Table 2 biology-11-00137-t002:** The comparison of secondary metabolites biosynthetic gene clusters in *B.*
*inaquosorum* strains KR2-7, *B. subtilis* 168 and *B.velezensis* strain FZB42.

Metabolite	Synthetase Type	Gene Cluster			Function	Gene Similarity with Strain KR2-7	
		*B. inaquosorum* KR2-7	*B. subtilis* 168	*B. velezensis* FZB42		168	FZB42
Bacillaene	PKS-NRPS	*pksABCDE*, *acpK*, *pksFGHIJLMNRS*	*pksABCDE*, *acpk*, *pksFGHIJLMNRS*	*baeBCDE*, *acpK*, *baeGHIJLMNRS*	Antibacterial	89.63%	75.25%
Bacillibactin	NRPS	*dhbABCEF*	*dhbABCEF*	*dhbABCEF*	Nutrient uptake	92.25%	73.07%
Bacilysin	NRPS	*bacABCDE*, *ywfAG*	*bacABCDEFG, ywfA*	*bacABCDE*, *ywfAG*	Antibacterial	93.50%	80.67%
Fengycin	NRPS	*ppsABCDE*	*ppsABCDE*	*fenEABCD*	Antifungal	92.01%	72.05%
Macrolactin	PKS	*pksJL*, *baeE*	-	*pks2ABCDEFGHI*	Antibacterial	-	74.12%
Bacillomycin F	NRPS	*ituD, ituABC*	-	-	Antifungal, ISR	-	-
Sporulation killing factor	Head-to-tail cyclised peptide	*skfABCEFGH*	*skfABCEFGH*	-	Antibacterial	96.08%	-
Subtilosin A	Thiopeptide	*sboA*, *albABCDEFG*	*sboA*, *albABCDEFG*	-	Antibacterial	91.86%	-
Surfactin	NRPS	*srfAA*, *AB*, *AC*, *AD*	*srfAA*, *AB*, *AC*, *AD*	*srfAA*, *AB*, *AC*, *AD*	Antifungal, Antibacterial, Colonization, ISR	92.20%	74.65%

**Table 3 biology-11-00137-t003:** Assignments of all fengycin mass peaks obtained by MALDI-TOF mass spectrometry of whole cells of strain KR2-7 grown on control and dual culture plates.

	Mass Peak (m/z)	Assignment	Reference
**On PDA control**	1501.9	Ala-6-C16 fengycin [M + H, Na, K]^+^	[38]
	1515.9	Ala-6-C17 fengycin [M + H, Na, K]^+^	[38]
	1529.9	Val-6-C16 fengycin [M + H, Na, K]^+^	[38]
	1543.8	Val-6-C17 fengycin [M + H, Na, K]^+^	[38]
**On PDA dual culture**	1471.9	Ala-6-C15 fengycin [M + H, Na, K]^+^	[38]
	1485.7	C16-Fengycin A [M + Na]	[39]
	1487.9	Ala-6-C15 fengycin [M + H, Na, K]^+^	[38]
	1499.9	Ala-6-C17 fengycin [M + H, Na, K]^+^	[38]
	1501.9	Ala-6-C16 fengycin [M + H, Na, K]^+^	[38]
	1513.8	C18-fengycin A [M + Na]	[39]
	1515.9	Ala-6-C17 fengycin [M + H, Na, K]^+^	[38]
	1527.8	Val-6-C17 fengycin [M + H, Na, K]^+^	[38]
	1529.9	Val-6-C16 fengycin [M + H, Na, K]^+^	[38]
	1543.8	Val-6-C17 fengycin [M + H, Na, K]^+^	[38]
	1501.9	Ala-6-C16 fengycin [M + H, Na, K]^+^	[38]
	1515.9	Ala-6-C17 fengycin [M + H, Na, K]^+^	[38]
	1527.8	Val-6-C16 fengycin [M + H, Na, K]^+^	[38]
	1529.9	Val-6-C17 fengycin [M + H, Na, K]^+^	[38]
	1543.8	Ala-6-C15 fengycin [M + H, Na, K]^+^	[38]

**Table 4 biology-11-00137-t004:** Assignments of iturin mass peaks obtained by MALDI-TOF mass spectrometry of whole cells of strain KR2-7 grown on control and dual culture plates.

	Mass Peak (m/z)	Assignment	Reference
**On PDA control**	1106.6	C17-iturin [M + Na]^+^	[18]
	1122.6	C17-iturin [M + K]^+^	[18]
	1134.6	C19-iturin [M + Na]^+^	[18]
	1136.6	C18-iturin [M + K]^+^	[18]
**On PDA dual culture**	1092.6	C16-iturin [M + Na]^+^	[18]
	1098.6	C16-iturin [M + H]^+^	[18]
	1106.6	C17-iturin [M + Na]^+^	[18]
	1112.6	C19-iturin [M + H]^+^	[18]
	1120.6	C18-iturin [M + Na]^+^	[18]
	1122.6	C17-iturin [M + K]^+^	[18]
	1134.6	C19-iturin [M + Na]^+^	[18]
	1136.6	C18-iturin [M + K]^+^	[18]
	1150.6	C19-iturin [M + K]^+^	[18]

**Table 5 biology-11-00137-t005:** Assignments of surfactin mass peaks obtained by MALDI-TOF mass spectrometry of whole cells of strain KR2-7 grown on control and dual culture plates.

	Mass Peak (m/z)	Assignment	Reference
**On PDA control**	1044.6	C14-surfactin [M + Na, K]^+^	[18]
	1046.6	C13-surfactin [M + K]^+^	[18]
	1058.6	C15-surfactin [M + Na]^+^	[18]
	1060.5	C14-surfactin [M + Na, K]^+^	[18]
	1074.6	C15-surfactin [M + Na, K]^+^	[18]
**On PDA dual culture**	1030.6	C13-surfactin [M + Na]^+^	[18]
	1032.7	C13-surfactin [M + K]^+^	[18]
	1044.6	C14-surfactin [M + Na, K]^+^	[18]
	1046.6	C13-surfactin [M + K]^+^	[18]
	1058.6	C15-surfactin [M + Na]^+^	[18]
	1060.5	C14-surfactin [M + Na, K]^+^	[18]
	1074.6	C15-surfactin [M + Na, K]^+^	[18]

**Table 6 biology-11-00137-t006:** Genes and gene clusters involved in plant-bacterium interaction in the genome of KR2-7.

Bioactivity	Gene/Gene Cluster	From	To	Product	Remark
Root colonization	*yfi*Q	1181545	1180457	Putative membrane-bound acyltransferase YfiQ	Involved in surface adhesion [13,84]
	*sac*B	2852367	2850946	Levan sucrase	Levan contributed to the aggregation of wheat root-adhering soil [85]
Swarming motility	*swr*B	319026	318514	Swarming motility protein swrB	Essential for swarming motility [86]
	*swr*C	1348113	1344916	Swarming motility protein SwrC	Self-resistance to surfactin [86]
	*sfp*	1712320	1712994	4’-phosphopantetheinyl transferase sfp	Necessary for lipopeptide and polyketide synthesis which is essential for surface motility and biofilm formation [13,86]
	*swr*AA	2770348	2770776	Swarming motility protein swrAA	Essential for swarming motility [87]
	*swr*AB	2770855	2772051	Swarming motility protein swrAB	Essential for swarming motility [87]
	*efp*	3829945	3830526	Elongation factor P	Essential for swarming motility [88]
	*flh*A	328577	326544	Flagellar biosynthesis protein flhA	Flagellar assembly
	*flh*B	329692	328610	Flagellar biosynthetic protein flhB	
	*fli*R	330489	329692	Flagellar biosynthetic protein fliR	
	*fli*Q	330757	330479	Flagellar biosynthetic protein FliQ	
	*fli*P	331428	330763	Flagellar biosynthetic protein fliP	
	*fli*Y	333625	332483	Flagellar motor switch phosphatase FliY	
	*fli*M	334613	333606	Flagellar motor switch protein FliM	
	*fli*L	335060	334638	Flagellar protein FliL	
	*flg*G	336163	335312	Flagellar basal-body rod protein flgG	
	*fli*K	338009	336546	Probable flagellar hook-length control protein	
	*fli*J	339085	338642	Flagellar FliJ protein	
	*fli*I	340404	339088	Flagellum-specific ATP synthase	
	*fli*H	341153	340401	Probable flagellar assembly protein fliH	
	*fli*G	342162	341146	Flagellar motor switch protein FliG	
	*fliF*	343785	342175	Flagellar M-ring protein	
	*fli*E	344151	343831	Flagellar hook-basal body complex protein FliE	
	*flg*C	344618	344163	Flagellar basal-body rod protein flgC	
	*flg*B	345010	344615	Flagellar basal-body rod protein flgB	
	*mot*A	654985	655887	Motility protein A	
	*mot*B	655835	656650	Motility protein B	
Swarming motility	*flh*O	2629891	2630742	Flagellar hook-basal body complex protein flhO	Flagellar assembly
	*flh*P	2630949	2631584	Flagellar hook-basal body complex protein flhP	
	*flg*M	2751851	2752117	Negative regulator of flagellin synthesis	
	*flg*K	2752634	2754151	Flagellar hook-associated protein 1	
	*flg*L	2754161	2755057	Flagellar hook-associated protein 3	
	*hag*	2756494	2757405	Flagellin	
	*fli*D	2757987	2759483	Flagellar hook-associated protein 2	
	*fli*S	2759505	2759906	Flagellar protein fliS	
	*fli*T	2759903	2760244	Flagellar protein FliT	
	*che*D	320333	319833	Chemoreceptor glutamine deamidase CheD	Bacterial chemotaxi
	*che*C	320959	320330	CheY-P phosphatase CheC	
	*che*W	321457	320978	Chemotaxis protein CheW	
	*che*A	323488	321470	Chemotaxis protein CheA	
	*che*B	324555	323485	Chemotaxis response regulator protein-glutamate methylesterase	
	*che*Y	332463	332095	Chemotaxis protein CheY	
	*che*V	616969	616058	Chemotaxis protein CheV	
	*che*R	3987238	3988155	Chemotaxis protein methyltransferase	
Biofilm formation	*pgs*A	273607	273026	CDP-diacylglycerol-glycerol-3-phosphate3-Phosphatidyl transferase	Member of *pgsB-pgsC-pgsA-pgsE* gene cluster encoding PGA which is contributed to robustness and complex morphology of the colony biofilms [89]
	*pgc*A	1083817	1082072	Phosphoglucomutase	Phosphoglucomutase plays an important role in biofilm formation [90]
	*ybd*K	1903365	1902379	Sensor histidine kinase ybdK	Transcriptional regulation of biofilm formation [91,92]
	*sig*W	1928876	1928313	RNA polymerase sigma factor sigW	Transcriptional regulation of biofilm formation [91,92]
	*sig*H	2011463	2010807	RNA polymerase sigma-H factor	Involves in the initial stage of biofilm formation [93]
	*abr*B	2082858	2083148	Transition state regulatory protein AbrB	Transcriptional regulation of biofilm formation [94]
	*eps*C-O	2731363	2874940	Gene cluster for capsular poly-saccharide biosynthesis	Encoding exopolysaccharide which is essential for biofilm formation [91]
	*lyt*S	3426013	3427803	Sensor protein lytS	Transcriptional regulation of biofilm formation [91,92]
	*yqx*M	3813390	3814151	Protein yqxM	Belongs to yqxM-sipW-tasA gene cluster that is necessary for biofilm formation [95]
	*tas*A	3814771	3815556	Spore coat-associated protein N	Required for development of complex colony architecture [94]
	*sin*R	3816019	3815651	HTH-type transcriptional regulator sinR	Transcriptional regulation of biofilm formation [91,92]
	*sin*I	3816289	3816020	Protein sinI	Transcriptional regulation of biofilm formation [91,92]
Biofilm formation	*spo*0A	3849786	3850625	Stage 0 sporulation protein A	Involved in the initial stage of biofilm formation [93]
	*res*E	3951273	3953042	Sensor histidine kinase resE	Transcriptional regulation of biofilm formation [91,92]
	*ymc*A	261856	261425	Protein ymcA	These genes are involved in the development of multicellular communities [91]
	*ylb*F	467892	467434	Regulatory protein ylbF	
	*yqe*K	3725618	3726187	Protein yqeK	
	*sip*W	3814123	3814707	Signal peptidase I W	
Mineral assimilation	*moa*D	594413	594180	Molybdopterin synthase sulfur carrier subunit	Nitrogen assimilation
	*moa*E	594879	594406	Molybdopterin synthase catalytic subunit	
	*moa*C	1469542	1469000	Molybdenum cofactor biosynthesis protein C	
	*moa*A	2592667	2593692	Molybdenum cofactor biosynthesis protein A	
	*moa*B	3369200	3369751	Molybdenum cofactor biosynthesis protein B	
	*nas*A-F	1758586	1768809	Gene cluster for Nitrate transport and reduction	Nitrogen assimilation
	*nar*K	2532129	2533373	Nitrite extrusion protein	
	*fnr*	2533450	2534190	Anaerobic regulatory protein	
	*arf*M	2534994	2535563	Probable transcription regulator arfM	
	*nar*G-J	2535783	2542171	Gene cluster for Nitrate reductase	
	*nrg*B	2619035	2618670	Nitrogen regulatory PII-like protein	
	*nrg*A	2620246	2619032	Ammonium transporter nrgA	
	*mgt*E	694316	692961	Magnesium transporter mgtE	Magnesium assimilation
	*cor*A	3806910	3807869	Magnesium transport protein CorA	
	*mnt*H	1626982	1628259	Manganese transport protein mntH	Manganese assimilation
	*mnt*A-D	3237914	3241819	Gene cluster for Manganese binding/transport protein	
	*mnt*R	3824735	3825202	Transcriptional regulator mntR	
	*ktr*C	573312	572647	Ktr system potassium uptake protein C	Potassium assimilation
	*ykq*A	574212	573469	Putative gamma-glutamylcyclo transferase ykqA	
	*ktr*D	674275	672926	Ktr system potassium uptake protein D	
	*yug*O	3174241	3173258	Putative potassium channel protein yugO	
	*ktr*B	3203187	3201850	Ktr system potassium uptake protein B	
	*ktr*A	3203862	3203194	Ktr system potassium uptake protein A	
Mineral assimilation	*ycl*Q	1683673	1682711	Ferrichrome ABC transporter	Iron assimilation
	*yvr*C	2985220	2986197	Putative iron binding lipoprotein yvrC	
	*yus*V	3009672	3010583	Putative iron (III) ABC transport ATPase component	
	*dhb*ABCEF	3101063	3112861	Gene cluster encoding Bacillibactin	
	*tua*A-H	2733487	2742546	Gene cluster for teichuronic acid biosynthesis	Bivalent cations assimilation
	*yvd*K	2839830	2842166	Glycosyl hydrolase yvdK	Ferrochrome assimilation
Plant growth promotion/ISR	*als*R	2672398	2671511	HTH-type transcriptional regulator alsR	These genes encode enzymes of the biosynthetic pathway from pyruvate to 3-hydroxy-2-butanone
	*als*S	2672549	2674270	Acetolactate synthase	
	*als*D	2674320	2675099	Alpha-acetolactate decarboxylase	
	*bdh*A	1448068	1449108	(R, R)-butanediol dehydrogenase	This gene encodes enzyme to acatlyse 3-hydroxy-2-butanone to 2,3-butanediol
Plant growth promotion	*yhc*X	1092936	1091395	Carbon-nitrogen hydrolase	These genes are involved in indole acetic acid biosynthesis
	*ysn*E	3489604	3489101	N-acetyltransferase	
	*dha*S	4136609	4135263	Putative aldehyde dehydrogenase dhaS	
	*phy*	4084788	4085936	3-phytase	Phytase hormones biosynthesis gene
	*tre*R	1237083	1236367	Trehalose gene cluster transcriptional repressor	These genes are involved in trehalose biosynthesis
	*tre*A	1238792	1237104	Trehalose-6-phosphate hydrolase	
	*tre*P	1240275	1238863	PTS system trehalose-specific EIIBC component	
	*ilv*H	3492606	3493124	Acetolactate synthase small subunit	These genes are parts of leucine, valine, and isoleucine biosynthesis pathway
	*ilv*B	3490858	3492609	Acetolactate synthase large subunit	
	*ilv*C	3493148	3494176	Ketol-acid reductoisomerase	
	*spe*A	501879	503375	Arginine decarboxylase	These genes may transform amino acids to plant growth-promoting substances [80]
	*spe*E	2512298	2513128	Spermidine synthase	
	*spe*B	2513189	2514061	Agmatinase	

## Data Availability

The genome sequence of B. inaquosorum KR2-7 was deposited in NCBI GenBank under the accession number QZDE00000000.2.

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
