# Peer review of "Perception of Biocontrol Potential of Bacillus inaquosorum KR2-7 against Tomato Fusarium Wilt through Merging Genome Mining with Chemical Analysis"

_biology, 2022, doi:10.3390/biology11010137_

Round 1
Reviewer 1 Report
Accept in present form
Author Response
Dear Reviewer,
Thanks a lot for your positive consideration.
Reviewer 2 Report
Interesting article and very well written. The methodology is advanced, but described in an accessible way, and the results are more than presented, they can even be shortened. The discussion is also interesting, and the conclusion is justified. Some minor comments below:
line 54: Bacillus in italics
line 87-89: I would delete this passage. The quality of the study and its usefulness will be assessed by the scientific community through quotations, there is no need to praise yourself
line 94-95: It is worth providing more details on the incubation of the strain, including conditions and time
line 144: The table must be positioned differently in the text. Its citation is earlier than Figure 1.
line 180: I would move the figure to the supplement, it is not essential to understand the results which are well described. However, this is only a suggestion to consider
line 189: Figure 2 is completely illegible. It must be reformatted. For due diligence, the species names should also be put in italics.
line 414-415 and the entire manuscript: It seems to me that the gene names are spelled incorrectly. Please check this
line 612: Table 6 is misplaced. The order of quoting and placing figures/tables should be kept
Author Response
Dear Reviewer,
Thanks for your useful comments.
According to your comments, the manuscript has been revised. Please see the attachment as our responses to your comments. Besides, the revised manuscript has been uploaded.

Reviewer 3 Report
The submitted paper is interesting. However, there some major problems which could solved (see below).
A1.The authors should in include in the introduction/ discussion and compare in detail the published work on Bacillus subtilis subsp. inaquosorum strain HU Biol-II and the type strain DE111.
A2.The ‘phylogenomic tree of strain KR2-7 and selected Bacillus strains’ gives very poor resolution. I would recommend a reconstruction of the phylogenomic tree using TYGS, and include Bacillus subtilis subsp. inaquosorum strain HU Biol-II and the type strain DE111.
B1.The values of genes similarity of 168 and FKZB2 with strain KR2-7 ( Table 2 ) should be re-examined and the authors should provide data and explanation how they did these calculations.Based on the antiSMASH analysis region 2.6 harbouring the KR2-7 surfactin BGC showed 93% of genes show similarity to KCTC 13429 (B. inaquosorum).
B2.The gene cluster of bacillibactin in FZB42 is declared as ‘dhbABCCEF’ and in 168 as ‘dhbABCEF’. Explain which one is the right one.
B3. The cluster of plipastatin/fengycin is usually referred as fenEABCD and no as fenABCDE
B4. In table 2 the word ‘Enzyme’ should be replaced (look at your antiSMASH analysis)
C1. Lines 231-232. The expression ‘The scattering of identified BGCs within KR2-7 genome underlies its vigorous potential in plant disease biocontrol application’, the word scattering is not appropriate.
D1. The authors state (Lines 527-530) ‘In addition to efficient root colonization, strain KR2-7 is able to directly suppress soil- 527 dwelling phytopathogens through the formation of protective zone in rhizosphere by diffusing eight antimicrobial secondary metabolites e.g. fengycin, surfactin, bacillomycin F, macrolactin, bacillaene, bacilysin, subtilosin A, and sporulation killing factor’. In this regard, they are pretty much aware where the metabolites should be located when cells are growing on a solid surface. However, the authors extracted the metabolites from cells within the colony (dual culture or singly grown on PDA) and not from the secreted and diffusible metabolites which are located infront of the singly grown colony or between the bacterial colony and the expanding fungal colony (dual culture.
Can they provide some justified explanation (and discuss it) for their choice?
D2. The authors state : This result indicated that strain KR2-7 produced different variants of iturins to limit the growth of Fol hyphae.refer to Bacillomycin F as a variant of Iturins ( Lines 299-300). Then they compare the m/z values of the ‘iturins’ identified in their studies to bacillomycin D (see Figure 5) and not to bacillomycin F identified by Dunlaps group. Then produce some data and they state (lines 621-623) ,’’Figure S2: The iturin, mycosubtilin, and surfactin mass peaks detected by MALDI-TOF mass spectrometry in A) control cell extract of KR2- 622 7 and B) dual culture cell extract of KR2-7’’.
The genomic data provided evidence that KR2-7 have the capacity to produce bacillomycin F, the chemical analysis of the ‘’ iturins’’ is not very clear. I would recommend to carefully clarify their data with respect to the production of ‘’ iturin, mycosubtilin, bacillomycin F’’ and present/discuss their identified their m/z values more carefully.
E1. Lines 511-513. .’The observation of m/z value 1120.6 511 in the MALDI-TOF-MS spectra of strain KR2-7 definitely showed that this strain is a B. inaquosorum’.
The authors should explain what they mean.
Author Response

(The authors gave the same response as above.)

Reviewer 4 Report
Dear authors,
Will the new biosecurity strategy for tomato with the strain B. inaquosorum KR2-7 be more effective than the physical and breeding methods used so far or the chemical treatment of plants with fungicides or the cultivation of resistant tomato varieties? Is the above bacterial strain an endophyte? If so, is it sufficient to apply it once to the leaves or soil of the rhizosphere, and once established, will it induce ISR resistance and support plant growth by facilitating the uptake of nutrients from the soil?
Do tested fungal pathogens that are repeatedly transplanted into media retain their pathogenicity unchanged, or do they need to be refreshed from time to time by isolating new pathogens from nature?
L78, to support your statement, I suggest adding a new publication on this topic https://doi.org/10.3390/f12121714 and in the discussion, if you mention about metabolites of B. amyloliquefaciens, you can cite https://doi.org/10.3389/fmicb.2017.01438.
L545 and further ...you can mention about protection of seedlings in forestry from oomycetes.
The author's contributions are in need of supplementation and Figure 12 could be a graphical abstract
Author Response
Dear Reviewer,
Thanks for your useful comments.
According to your comments, the manuscript has been revised. Please see the attachment as our responses to your comments. Besides, the revised manuscript has been uploaded

Round 2
Reviewer 3 Report
A1. In their response the authors state that they collected cells from the ‘’outer layer of bacterial colony’’(not within the colony). So they extracted metabolites which are still within the cells.
They should have collected agar infront of the colony or agar from the clear zone (dual culture). In the M an M section this should become clear. Also in the results and discussion section should be discussed.
A2. The illustration of fengycin BGC is wrong. The authors should reproduce the right one. Consult your antismash analysis.
The number 4232045-4255145 under the ppsA-ppsC what they mean. The whole genome contains about 4.107.000 bp.
For numbering the authors should have a look at their antismash analysis.
A2.1. The size of fengycin BGC identified by the authors is 32. 4 kB. The real size of plipastatin is 37.74 kB. They better compare their fengycin to that of B. inaquosorum strain HU Biol-II, and then make a comment on scores of gene similarity.
B1. The illustration of subtilosin BGC is wrong. Some genes are missing and the size of the BGC is not 3.6kB. The numbering above the cluster is not compatible to that given in the antismash analysis. Consult your antismash analysis for proper presentation and estimations of the size of all BGC. B1.1 The nubering under the surfactin BGC is wrong.
C1. Table 2. Most of the gene similarity scores are wrong.
For example, the bacilysin BGC share 93.69 and 80.47% (nucleotide identity) with the respective BGC in B. subtilis strain 168 and B. velezensis FZB42, respectively.
To my opinion the bioinformatics and antismash data as well as Table 2 should be re-examined and presented in a proper way.
At the present, this manuscript suffers from problems and do not merit to be considered for publication before these problems are solved.
Author Response
Dear reviewer,
Thanks for rigorously reviewing our manuscript. Please find our responses to your comments in the attached files.
